# Misophonia in the UK: Prevalence and norms from the S-Five in a UK representative sample

**Silia Vitoratou**[1]*, **Chloe Hayes**[1], **Nora Uglik-Marucha**[1], **Oliver Pearson**[1], **Tom Graham**[2], **Jane Gregory**[3,4]

1 Biostatistics and Health Informatics Department, Psychometrics and Measurement Lab, Institute of Psychiatry, Psychology and Neuroscience, King's College London, London, United Kingdom, 2 Centre for Anxiety Disorders and Trauma, South London and Maudsley NHS Foundation Trust, London, United Kingdom, 3 Department of Experimental Psychology, University of Oxford, Oxford, United Kingdom, 4 Oxford Health Specialist Psychological Interventions Centre, Oxford Health NHS Foundation Trust, Oxford, United Kingdom

* silia.vitoratou@kcl.ac.uk

**Data Availability Statement:** The data used in this study are available from https://reshare.ukdataservice.ac.uk/856149/ (Vitoratou, Silia (2023). Misophonia in the UK: Prevalence and

## Abstract

What is the reality of the misophonic experience in the general population? This is a study on misophonia in a large sample, representative of the UK general population. The study utilises a multidimensional psychometric tool, the S-Five, to study the intensity of the triggering misophonic sounds in everyday activities, the emotions/feelings related to them, and the norms of the key components of the misophonic experience: internalising and externalising appraisals, perceived threat and avoidance behaviours, outbursts, and the impact on functioning. Based on the S-Five scores and a semi-structured interview delivered by clinicians who specialise in misophonia, the estimated prevalence of people for whom symptoms of misophonia cause a significant burden in their life in the UK was estimated to be 18%. The psychometric properties of the S-Five in the UK general population were also evaluated and differences across gender and age were explored. Our results show that the five-factor structure is reproduced, and that the S-Five is a reliable and valid scale for the measurement of the severity of the misophonic experience in the general UK population.

## Introduction

Misophonia, by consensus [1], is recognised as a disorder characterised by a disproportionate emotional response to everyday sounds [2]. The misophonic response can range from mild irritation to anger and distress and can result in impairment to social and occupational functioning [3,4]. Common manifestations of misophonia include feelings of anger, disgust, and anxiety [4–8]; muscle tension [3,9]; avoidance of triggering stimuli [3,10–12], withdrawal from social situations [10–15] and, in some cases, verbal and physical aggression [6,10,13,16]. Secondary emotional responses have also been reported, for example shame, guilt [5,17] and anticipatory anxiety [10].

To date, there is little research on the prevalence of misophonia, with reported estimates varying between 5% and 20% in specific samples. A study of 483 American undergraduate

Norms for the S-Five in a UK Representative Sample, 2020-2022. [Data Collection]. Colchester, Essex: UK Data Service. 10.5255/UKDA-SN-856149).

**Funding:** SV, CH, OP and NUM were funded by the NIHR Maudsley Biomedical Research Centre at South London and Maudsley NHS Foundation Trust and King's College London. This research was funded in whole, or in part, by the Wellcome Trust [JG; Grant number 102176/B/13/Z]. For the purpose of open access, the author has applied a CC BY public copyright licence to any Author Accepted Manuscript version arising from this submission. The views expressed are those of the author(s) and not necessarily those of the NHS, The Wellcome Trust, the NIHR or the Department of Health and Social Care. The funders had no role in study design, data collection and analysis, decision to publish, or preparation of the manuscript. There was no additional external funding received for this study.

**Competing interests:** The authors have declared that no competing interests exist.

students, reported that nearly 20% of the sample experienced clinically significant symptoms of misophonia [18]. In a study of 415 Chinese university students reported that 20% of participants were either "often" or "always" sensitive to sounds of people eating, nasal sounds and repetitive tapping, and 6% reported clinically significant misophonia [19]. A study on the prevalence of misophonia in Turkey reported the prevalence of 12.8% while 78% of the participants reported to experience aversion to at least one sound [20]. Naylor, Caimino [21] found almost half of the undergraduate medical students of their sample reported clinically significant symptoms of misophonia, with the majority (37%) reporting mild symptoms of misophonia and a small number of students (0.3%) reporting severe misophonia. The higher prevalence rate in this population may be explained by the increased propensity of medical students to experience conditions which have been found to co-occur with misophonia.

Research studies have used self-report questionnaires to study misophonia in the general population [18,19,21] and in individuals identifying with the condition [3,4,6]. Vitoratou [22] used the responses of individuals who identified with having misophonia in four waves of sampling to develop a robust psychometric tool that assesses the severity of the misophonic experience. Their work resulted in the S-Five (Selective Sound Sensitivity Syndrome Scale) which surfaced five dimensions of the misophonic experience: a sense of emotional threat, internalising and externalising appraisals, outbursts and impact. In this work we use this scale to assess misophonia in the UK general population.

To date, no study has investigated misophonia in a sample representative of the UK general population. First, the factor structure and psychometric properties of the S-Five were assessed in this population. The S-Five was then used to present estimates of the intensity of the emotions caused by sounds which trigger misophonic reactions in everyday activities, and the norms related to the five key components of the misophonic experience. A third aim of the study was to estimate the prevalence of people in the UK for whom symptoms of misophonia cause a significant burden in their life. We aimed to use the outcomes of semi-structured interviews to determine the point on the S-Five at which someone can be considered likely to have significant symptoms of misophonia, and to use that to estimate its prevalence in the UK general population.

## Methods

### Recruitment

Participants constituted a representative sample of the UK general population, recruited via Prolific.co, via an allocation algorithm to stratify sample size across sex, age, and ethnicity using census data from the UK [23]. Participants read and consented to participants' information sheet (ethics approval reference RESCM-19/20-11826) and were subsequently screened for their eligibility criteria, which included being aged 18 years or older, English fluency, and no diagnosis of a severe learning or intellectual disability.

### Measures

An extended battery of 17 scales were considered within the S-Five study, described in Vitoratou [22] and reprinted here in the (S4 Table). In this section we present the tools used in the current validation.

**Selective sound sensitivity syndrome scale [S-Five; 22].** The S-Five consists of 25 items assessing the severity of misophonia. It is rated an interval scale from 0: not at all true to 10: completely true. The severity scale is also complemented by a trigger checklist (S-Five-T), to assess the trigger sounds, the response to them, and the intensity of the response. The checklist currently uses 37 triggers suggested by research data, but researchers can add or remove

sounds. The type of reaction to the trigger can be recorded and in this study were: no feeling, irritation, distress, disgust, anger, panic, other feeling: negative, and other feeling: positive. Each trigger item also rates the intensity (henceforth trigger intensity) of the reaction (from 0: doesn't bother me at all to 10: unbearable/causes suffering). This allows for the computation of four indices: 1) the trigger count (TC), which is the total number of triggers endorsed by a participant from the list provided, 2) the reaction count (RC), which is the number of times each particular reaction type is endorsed and can be counted across triggers in a single participant, or across participants, 3) the frequency/intensity of reactions score (FIRS) is the total value of the intensity items of all endorsed triggers, and 4) the relative intensity of reactions score (RIRS) which gives an estimate of the intensity of reactions to triggers, relative to the number of triggers reported. The 25 statements and the trigger checklist used in this study are reprinted here in the (S1 Table) along with the details and examples for the computation of the five factors and the four trigger indices, originally presented in Vitoratou [22,24].

**Misophonia Questionnaire [MQ; 18].** The MQ is a self-report measure consisting of three measures for misophonia: the Misophonia Symptom Scale (MSYS) which assesses sensitivity to specific triggers in comparison to other people, the Misophonia Emotions and Behaviours Scale (MEBS) which relates to an individual's reactions to triggering sounds. The MQ total score which is calculated by combining the scores of both the MSYS and the MEBS. The third section of the MQ, is the Misophonia Severity Scale [MSES; 18]. It is a single item which asks individuals to rate the severity of their sound sensitivity on a scale from 1 (minimal) to 15 (very severe), with a score greater than or equal to 7 indicating clinically significant symptoms.

**Amsterdam Misophonia Scale [A-MISO-S; 4].** The A-MISO-S was adapted from the Yale-Brown Obsessive-Compulsive Scale [YBOCS; 25]. The A-MISO-S address different aspects of misophonia, including time spent occupied by misophonia, impact on functioning, distress, attempts to resist, perceived control over sounds and thoughts, and avoidance. An interviewer discusses the questions with the patient and uses clinical judgement to rate each item [4], although it has also been used as a self-report tool [26].

**Patient Health Questionnaire-9 [PHQ-9; 27].** PHQ-9 has 9 items measuring the severity of depression with items scored on a 4-point ordinal scale, and a total score range of 0 to 27.

**General Anxiety Disorder-7 questionnaire [GAD-7; 28].** GAD-7 is a 7-item scale measuring severity of anxiety symptoms, rated on a 4-point ordinal scale and a total score ranging from 0 to 21.

**The diagnostic interview.** A preliminary version of the Oxford King's Structured Clinical Interview for Misophonia (Pre-OK-SCIM, in development by the authors) was used. The preliminary version used for the present study contained a series of questions and prompts to determine whether six key criteria were met, adapted from the Amsterdam UMC revised diagnostic criteria for misophonia [6]. The modifications were made based on outcomes from recent research [22,24] and observations in clinical practice. Notably, we did not require an oral or nasal sounds to be a trigger (A), intense reactions were not limited to irritation, anger and disgust (B), the individual did not need to recognise the excessive nature of the response (B), loss of control included experiencing panic and helplessness (C), and coping strategies were included (D).

An outcome of "significant misophonia" on the Pre-OK-SCIM indicated that the individual was significantly burdened by misophonia in their life at the time of the interview. It was not intended to be a clinical diagnosis of misophonia at a disorder level, and therefore does not assume levels of distress and impairment on par with, for instance, diagnoses such as obsessive-compulsive disorder or posttraumatic stress disorder. The Pre-OK-SCIM was administered by registered psychologists experienced with misophonia to allow for flexibility and clinical judgement in the use of the protocol.

## Statistical analysis

The latent structure of the S-Five was assessed using exploratory (EFA) and confirmatory factor analysis (CFA). The data were checked for their suitability for factor analysis using the Kaiser-Meyer-Olkin (KMO) test for sampling adequacy [29,30] and Bartlett [31] test of sphericity. In EFA, the maximum likelihood estimator with robust standard errors [MLR; 32] was incorporated due to data being skewed, with Oblimin rotation. To establish the number of factors, the Guttman-Kaiser criterion [29,33] and the parallel analysis criterion [34] were followed, depicted using Cattell's scree plot [35]. The percentage of variance explained was also evaluated [see for instance 36]. Goodness of fit indices were computed to assess the relative and absolute fit of competing models. The measures of fit that are reported include the relative chi-square [relative $\chi^2$: values close to 2 suggest an acceptable fit; 37], the Root Mean Square Error of Approximation [RMSEA: values < .06 are required for adequate fit; 38], the Taylor-Lewis Index [TLI: values >.95 suggest close fit; 38], the Comparative Fit Index [CFI: values >.95 are required for close fit; 38,39] and the Standardized Root Mean Residual [SRMR: values < .08 are needed for good fit; 38]. The multiple indicator multiple causes model [MIMIC; 40,41] was used to assess the measurement invariance in relation to gender and age.

Internal consistency was computed within each factor using the Cronbach's alpha coefficient [42; α] and McDonald's [43] Omega (ω). Test-retest reliability was assessed at item level by computing the Psi coefficient [44], to accommodate the skewness of the data on item level, and at factor level using the (mixed effects, absolute agreement) Intraclass Correlation Coefficient [ICC; 45]. The latter was evaluated according to interpretation guidelines outlined by Landis and Koch [46]. The assessment of convergent validity and hypothesis testing were conducted using parametric (Pearson's r, t-test) and non-parametric (Spearman's rho, Mann-Whitney test) methods depending on the distribution of the data.

To establish a cut-off score for significant misophonia from the S-Five, and subsequently estimate of prevalence, receiver operating characteristic (ROC) curve analysis was carried out. Using the outcome of the Pre-OK-SCIM, those with significant misophonia were classified as cases and those without this outcome were classified as controls. The ROC [47,48] curves were plotted for each of the S-Five subscales and total score and for the S-Five-T variables, with the Pre-OK-SCIM caseness as the classification variable. First, the area under the curve (AUC) values were considered, with good predictive ability achieved by an AUC above or equal to 0.8. Where an adequate AUC was established, the optimal cut-off scores were considered, for which a balance of sensitivity and specificity, close to 80%, and the [49] J index. The ROC analysis was extended to test the presence of significant covariates through ROC regression analysis [50–52].

The interview was conducted on a sample from the general population and a self-reporting misophonia sample. Thirty individuals who identified with the condition were randomly invited for the interview. A second sample of 30 individuals from the Prolific sample (representative of the UK population) were invited, which included individuals from all 10th-tiles of the total S-Five distribution, to ensure representation of the interviews of various levels of severity and to ensure the presence of people with significant misophonia.

Data analyses were conducted using MPlus 8 [32], Stata 16 [53], and R [54] statistical packages.

# Results

## Descriptive indices

With respect to gender, 396 individuals identified as females (7 trans women), 372 as males (1 trans man), and 4 identified as non-binary or other. The mean age was 46.4 years old (standard

deviation SD = 15.5, min = 19, max = 83) and did not differ across genders (t = 0.905, df = 758, p = 0.366). Only 13.6% of the sample was aware of the term misophonia and 2.3% identify as having the disorder.

## S-Five statements

**Statement responses.** The norms of each of the S-Five item/statements for the UK population are presented in Table 1. More highly endorsed were the item statements which refer to 'externalising appraisals' (for example I06 'others should avoid making noises'). The least endorsed statements were the statements related to being verbally aggressive (I04) and violent (I24), and impact (I01 'do not meet friends' and I20 'limited job opportunities').

**Table 1. Descriptive indices, associations with age and gender, factor analysis loadings, and reliability indices of the 25 S-Five items (N = 772).**

| S-Five-E statements per factor | mean (sd) | median (Q1-Q3) | mode (min-max) | Age rho | Gender difference mean (se)‡ | loadings EFA (CFA) [a] | Psi (95% CI) | ICC |
|---|---|---|---|---|---|---|---|---|
| **Externalising** | | | | | | | | |
| I06 Others avoid making noises | 4.8 (3.3) | 5 (2–8) | 0 (0–10) | **0.11 | 0.40 (0.2) | 0.67 (0.73) | 0.81 (0.8,1) | 0.86 |
| I13 Others should not make sounds | 4.1 (3.3) | 4 (1–7) | 0 (0–10) | 0.06 | 0.35 (0.2) | 0.82 (0.76) | 0.80 (0.8,1) | 0.86 |
| I16 Others selfish | 2.9 (3.0) | 2 (0–5) | 0 (0–10) | -0.04 | 0.36 (0.2) | 0.69 (0.82) | 0.80 (0.8,1) | 0.86 |
| I21 Others bad manners | 4.9 (3.3) | 5 (2–8) | 10 (0–10) | -0.03 | 0.12 (0.2) | 0.94 (0.76) | 0.79 (0.8,1) | 0.85 |
| I25 Others disrespectful | 2.7 (3.0) | 1 (0–5) | 0 (0–10) | -0.01 | 0.38 (0.2) | 0.67 (0.73) | 0.80 (0.8,1) | 0.86 |
| **Internalising** | | | | | | | | |
| I05 Respect myself less | 0.9 (2.0) | 0 (0–1) | 0 (0–10) | **-0.12 | -0.01 (0.1) | 0.83 (0.79) | 0.75 (0.7,1) | 0.84 |
| I08 Unlikeable person | 1.2 (2.1) | 0 (0–1) | 0 (0–10) | **-0.22 | -0.21 (0.2) | 0.89 (0.85) | 0.77 (0.7,1) | 0.85 |
| I12 Angry person inside | 1.6 (2.5) | 0 (0–2) | 0 (0–10) | **-0.17 | -0.21 (0.2) | 0.82 (0.86) | 0.78 (0.8,1) | 0.85 |
| I18 Bad person inside | 1.2 (2.2) | 0 (0–1) | 0 (0–10) | **-0.24 | -0.29 (0.2) | 0.66 (0.87) | 0.77 (0.7,1) | 0.85 |
| I19 Dislike self | 1.4 (2.4) | 0 (0–2) | 0 (0–10) | **-0.20 | -0.07 (0.2) | 0.82 (0.83) | 0.78 (0.8,1) | 0.85 |
| **Impact** | | | | | | | | |
| I01 Do not meet friends | 0.6 (1.4) | 0 (0–1) | 0 (0–10) | **-0.12 | 0.05 (0.1) | 0.55 (0.61) | 0.75 (0.7,1) | 0.84 |
| I09 Eventually isolated | 0.9 (1.9) | 0 (0–1) | 0 (0–10) | **-0.15 | -0.09 (0.1) | 0.71 (0.91) | 0.76 (0.7,1) | 0.84 |
| I14 Avoid places | 1.3 (2.3) | 0 (0–1) | 0 (0–10) | -0.05 | -0.03 (0.2) | 0.66 (0.67) | 0.76 (0.7,1) | 0.84 |
| I15 Cannot do everyday things | 0.8 (1.7) | 0 (0–1) | 0 (0–10) | **-0.10 | -0.06 (0.1) | 0.60 (0.91) | 0.75 (0.7,1) | 0.84 |
| I20 Limited job opportunities | 0.5 (1.4) | 0 (0–0) | 0 (0–10) | **-0.14 | -0.13 (0.1) | 0.38 (0.68) | 0.74 (0.7,1) | 0.84 |
| **Outburst** | | | | | | | | |
| I04 Verbally aggressive | 1.7 (2.4) | 1 (0–2) | 0 (0–10) | **-0.22 | -0.12 (0.2) | 0.55 (0.78) | 0.78 (0.8,1) | 0.85 |
| I17 Physically aggressive | 0.8 (1.6) | 0 (0–1) | 0 (0–10) | -0.07 | *0.21 (0.1) | 0.73 (0.77) | 0.74 (0.7,1) | 0.84 |
| I22 Violence | 0.5 (1.3) | 0 (0–0) | 0 (0–9) | **-0.14 | 0.13 (0.1) | 0.56 (0.72) | 0.71 (0.7,1) | 0.83 |
| I23 Shout at people | 1.8 (2.6) | 1 (0–3) | 0 (0–10) | **-0.12 | -0.01 (0.2) | 0.65 (0.73) | 0.79 (0.8,1) | 0.85 |
| I24 Afraid of outburst | 0.6 (1.5) | 0 (0–1) | 0 (0–9) | **-0.13 | 0.13 (0.1) | 0.56 (0.69) | 0.73 (0.7,1) | 0.83 |
| **Threat** | | | | | | | | |
| I02 Panic or explode | 2.3 (2.9) | 1 (0–4) | 0 (0–10) | **-0.18 | -0.39 (0.2) | 0.73 (0.82) | 0.82 (0.8,1) | 0.86 |
| I03 Feel helpless | 2.2 (2.9) | 1 (0–4) | 0 (0–10) | **-0.14 | -0.26 (0.2) | 0.80 (0.87) | 0.82 (0.8,1) | 0.86 |
| I07 Feel anxious | 3.0 (3.3) | 2 (0–5) | 0 (0–10) | **-0.14 | -0.42 (0.2) | 0.66 (0.88) | 0.84 (0.8,1) | 0.87 |
| I10 Experience distress | 4.0 (3.3) | 3 (1–7) | 0 (0–10) | **-0.14 | -0.11 (0.2) | 0.77 (0.79) | 0.84 (0.8,1) | 0.87 |
| I11 Feel trapped | 3.1 (3.3) | 2 (0–6) | 0 (0–10) | **-0.10 | -0.40 (0.2) | 0.60 (0.87) | 0.83 (0.8,1) | 0.87 |

Q1 Q3 first and third quartile; ICC intraclass correlation coefficient; Psi coefficient and 95% confidence intervals; rho: Spearman's correlation coefficient; *p<0.05

**p<0.01

‡ mean difference (se) male vs female comparison, p-value via Mann Whitney test; [a]CFA loadings were standardised (STDXY).

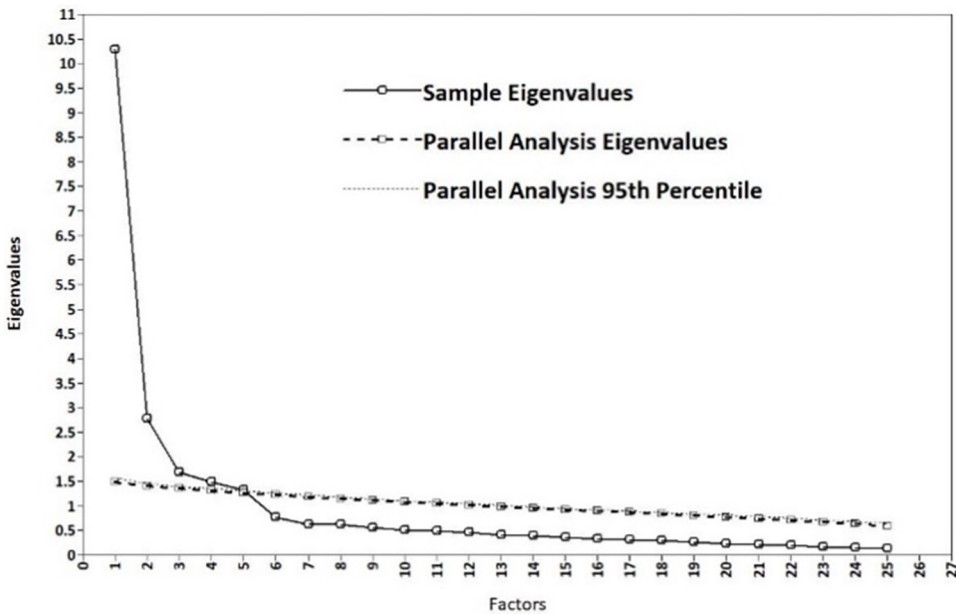

**Fig 1. Scree plot and parallel analysis visualisation.**

With respect to reported gender, females scored significantly lower than males in one of the statements (I17 'physically aggressive'; Table 1). Almost all items had significant but weak, negative correlations with age. The older the responder, the lower the endorsement of the statements were, apart from I13 'others should not make sounds' which was weakly related to age.

## EFA and CFA: Dimensionality and measurement invariance

The data were suitable for factor analysis according to the diagnostic criteria (anti-image correlations >0.88 in all statements, KMO = 0.94, Bartlett's test: $\chi^2$ = 13773,1, df = 300, p<0.001).

The first random split half of the data was used in EFA. The sample correlation matrix with five eigenvalues above 1 (10.3, 2.8, 1.7, 1.5, and 1.3) suggested a five-factor structure according to Kaiser-Guttman criterion, explaining 73% of the total variance. Parallel analysis also indicated that five factors should be extracted, as is depicted in the Scree plot in Fig 1. The intercorrelations among the five factors ranged from 0.151 (externalising-impact) to 0.512 (externalising-threat). The goodness of fit examination suggested adequate to close fit indices for the five-factor model (rel $\chi^2$ = 2.25; RMSEA = 0.057 with 95% (0.050,0.065), TLI = 0.90, CFI = 0.93, SRMR = 0.025) with all items loading above 0.4 on their respective factors and no cross-loadings present. We therefore accepted this solution (Table 2) and proceeded with CFA in the second split half of the data which also indicated good absolute fit and acceptable relative fit to our data (rel $\chi^2$ = 2.38; RMSEA = 0.066 with 95% (0.060,0.072), TLI = 0.88, CFI = 0.89, SRMR = 0.062). The model presented in Table 2 (EFA and CFA loadings) coincides with the original model found by Vitoratou [22] using responses from people identifying with misophonia.

The complete sample was used in the evaluation of the measurement invariance of the tool with respect to gender and age using a MIMIC model. At least one item per factor was found to be non-invariant due to gender (six significant direct effects d.e. in total: I02 d.e. = 0.22, I04 d.e. = 0.23, I05 d.e. = 0.35, I12 d.e. = 0.48, I14 d.e. 0.12, I20 d.e = 0.47, and I25 d.e. = -0.61). However, the actual differences in the expected scores were remarkably low, as the magnitude

**Table 2. Norms and reliability of the S-Five 5 factors and total scores (N = 772).**

| Factor | descriptive indices | | | | | internal consistency | | stability | |
|---|---|---|---|---|---|---|---|---|---|
| | mean (sd) | median (Q1-Q3) | mode (min-max) | Gender difference mean (sd)‡ | Age rho | α / ω | ITC | Psi (95% CI) | ICC |
| Externalising | 19.38 (12.9) | 18 (9–29) | 7 (0–50) | -1.6 (0.9) | 0.02 | 0.87 / 0.87 | 0.66–0.73 | 0.84 (0.8,1) | 0.87 |
| Internalising | 6.34 (9.7) | 2 (0–9) | 0 (0–49) | 0.8 (0.7) | **-0.23 | 0.91 / 0.92 | 0.70–0.84 | 0.82 (0.8,1) | 0.86 |
| Impact | 4.05 (7.0) | 1 (0–5) | 0 (0–48) | 0.3 (0.5) | **-0.11 | 0.85 / 0.86 | 0.56–0.79 | 0.81 (0.8,1) | 0.86 |
| Outburst | 5.28 (7.5) | 2 (0–7) | 0 (0–44) | -0.3 (0.5) | **-0.17 | 0.83 / 0.84 | 0.61–0.71 | 0.81 (0.8,1) | 0.86 |
| Threat | 14.58 (13.8) | 10 (3–23) | 0 (0–50) | 1.6 (1.0) | **-0.17 | 0.93 / 0.93 | 0.75–0.85 | 0.87 (0.8,1) | 0.89 |
| S-Five total | 49.64 (40.1) | 38 (18–72) | 0 (0–215) | 0.7 (2.9) | **-0.13 | 0.94 / 0.94 | 0.43–0.75 | 0.87 (0.8,1) | 0.88 |

‡ mean difference (standard error) male vs female comparison, p-value via Mann Whitney test; sd: Standard deviation; Q1 and Q3 first and third quartile respectively; α: Cronbach's alpha; ω: McDonald's omega; ITC: Item-total correlations; ICC: Intraclass correlation coefficient (two-way mixed effects, absolute agreement).

of all gender direct effects found significant was half a unit over eleven possible units. For example, for the same levels of latent sound sensitivity, women are expected to score significantly higher on the internalising statement I05 'respect myself less' by 0.35 units, on the 0–10 scale. When it comes to age, the effects were even less, with about 0.03 units expected increase per year of age (I05 d.e. = 0.03 and I08 d.e. = 0.02). Therefore, we consider the bias introduced in the measurement by gender and age minimal if not negligible and we conclude that the assessment of structural invariance is attainable.

## Norms for the UK population, reliability, and validity

The norms of the S-Five factor scores for the UK general population are presented in Table 2. There were no significant differences in the total and factor scores with respect to gender. All S-Five scores were significantly lower than those reported in the Vitoratou [22] sample of individuals who identify with the condition (p<0.001 in all cases; data available on request).

With respect to internal consistency, alpha and omega were satisfactory for all factors (0.83 or higher; Table 2), while test-retest reliability was also satisfactory with ICC being larger than 0.86 for all S-Five scores. No significant differences occurred with respect to gender for each factor of the S-Five, while negative low correlations emerged with age (-0.12 to -0.20), except for the externalising factor (Table 2). The factor intercorrelations were moderate to moderately strong and positive, as anticipated (Table 3).

Next, we follow Vitoratou [22] and we present the correlations of the S-Five factor and total scores with the two other misophonia scales (MQ and A-MISO-R), PHQ9 (depression) and GAD7 (anxiety; Table 3). All S-Five scores emerged moderately strong, positive significant correlations with the MQ and A-MISO-R scores, thus providing evidence of the concurrent, convergent validity of the measurements. Correlations between the S-Five total scores and PHQ9 and GAD7 were weak to moderate with the lowest correlation between the externalising factor and PHQ9, followed by correlations between GAD7 and externalising factor, and PHQ9 and the impact factor.

## S-Five trigger checklist (S-Five-T)

The scoring guide and the programming codes (SPSS, R project, Stata) to obtain factors and indices are freely available upon request made to the first author.

**Trigger count and reactions per trigger.** On average, individuals reported a negative reaction to 17 triggers out of 37. Only 28 individuals selected no feeling to all sounds presented.

**Table 3. Intercorrelations of the S-Five scores, and correlations with other measures (validity).**

|  | Externalising | Internalising | Impact | Outburst | Threat | Total S-Five |
|---|---|---|---|---|---|---|
| **S-Five (N = 772)** | | | | | | |
| Internalising | 0.39 | | | | | |
| Impact | 0.36 | 0.61 | | | | |
| Outburst | 0.48 | 0.61 | 0.56 | | | |
| Threat | 0.54 | 0.62 | 0.60 | 0.60 | | |
| Total | 0.79 | 0.73 | 0.67 | 0.73 | 0.88 | |
| **A-MISO-S (N = 396)** | | | | | | |
| Total | 0.38 | 0.47 | 0.47 | 0.43 | 0.55 | 0.59 |
| **MQ (N = 376)** | | | | | | |
| MSYS (N = 295) | 0.38 | 0.41 | 0.30 | 0.39 | 0.44 | 0.49 |
| MEBS (N = 286) | 0.50 | 0.53 | 0.51 | 0.62 | 0.68 | 0.72 |
| MSES (N = 376) | 0.37 | 0.45 | 0.44 | 0.38 | 0.56 | 0.56 |
| Total (N = 295) | 0.51 | 0.55 | 0.47 | 0.60 | 0.66 | 0.71 |
| **PHQ9 (N = 761)** | | | | | | |
| Total | 0.19 | 0.34 | 0.27 | 0.29 | 0.40 | 0.37 |
| **GAD7 (N = 772)** | | | | | | |
| Total | 0.25 | 0.37 | 0.31 | 0.30 | 0.45 | 0.43 |

Correlations are Spearman's rho and p-value<0.01 in all cases; A-MISO-S: Amsterdam Misophonia Scale; MQ: Misophonia Questionnaire; MSYS: Misophonia Symptoms Scale; MEBS: Misophonia Emotions and Behaviours Scale; MSES: Misophonia Severity Scale; PHQ-9: Physical Health Questionnaire; GAD-7: Generalised Anxiety Disorder Assessment.

For each specific sound we computed the percentage of individuals who selected each reaction. In this general population sample, the no feeling option was selected by most of the participants across all sounds (see Fig 2 which presents the percentage of respondents which selected each reaction for the 37 trigger items; for example, for the trigger sound 'yawning', 84% of the participants selected 'no feeling', 12% 'irritation', 1% selected 'distress', etc).

For sounds such as 'normal breathing', 'yawning', 'footsteps', and 'certain accents (letters)', more than 80% of the participants reported no feeling. On the contrary, there were sounds where the percentages were reversed. For instance, less than 25% percent of the participants reported no feeling when it came to the sounds 'teeth sucking', '[dog] barking', 'slurping', 'chewing gum', 'snoring', 'sniffing', 'coughing', and 'loud breathing'. The most frequently reported negative reaction was irritation, for all trigger sounds except loud chewing, for which disgust was more frequently reported (39%). The largest percentage of individuals reporting distress was in relation to 'baby crying' (21%), reporting anger was in relation to 'snoring' (15%) and 'barking' (14%), and reporting panic was in relation to 'footsteps' (4%).

**Reaction counts.** For each specific reaction we computed the number of times (that is number of trigger sounds) it was selected, over all triggers, to compute the corresponding reaction count (RC).

The norms for the UK population of the RC for each reaction to trigger sounds are presented in Table 4. Women related more often than men to feelings of disgust and distress to the triggers sounds, whereas men reported more often no feeling related to the triggers. With respect to age, there were significant low negative correlations with all RC apart from irritation where the correlation was positive (Table 4).

Table 5 presents the Intercorrelations of the S-Five, S-Five-T scores, and correlations with other measures. There were significant positive moderate correlations between RCs and the

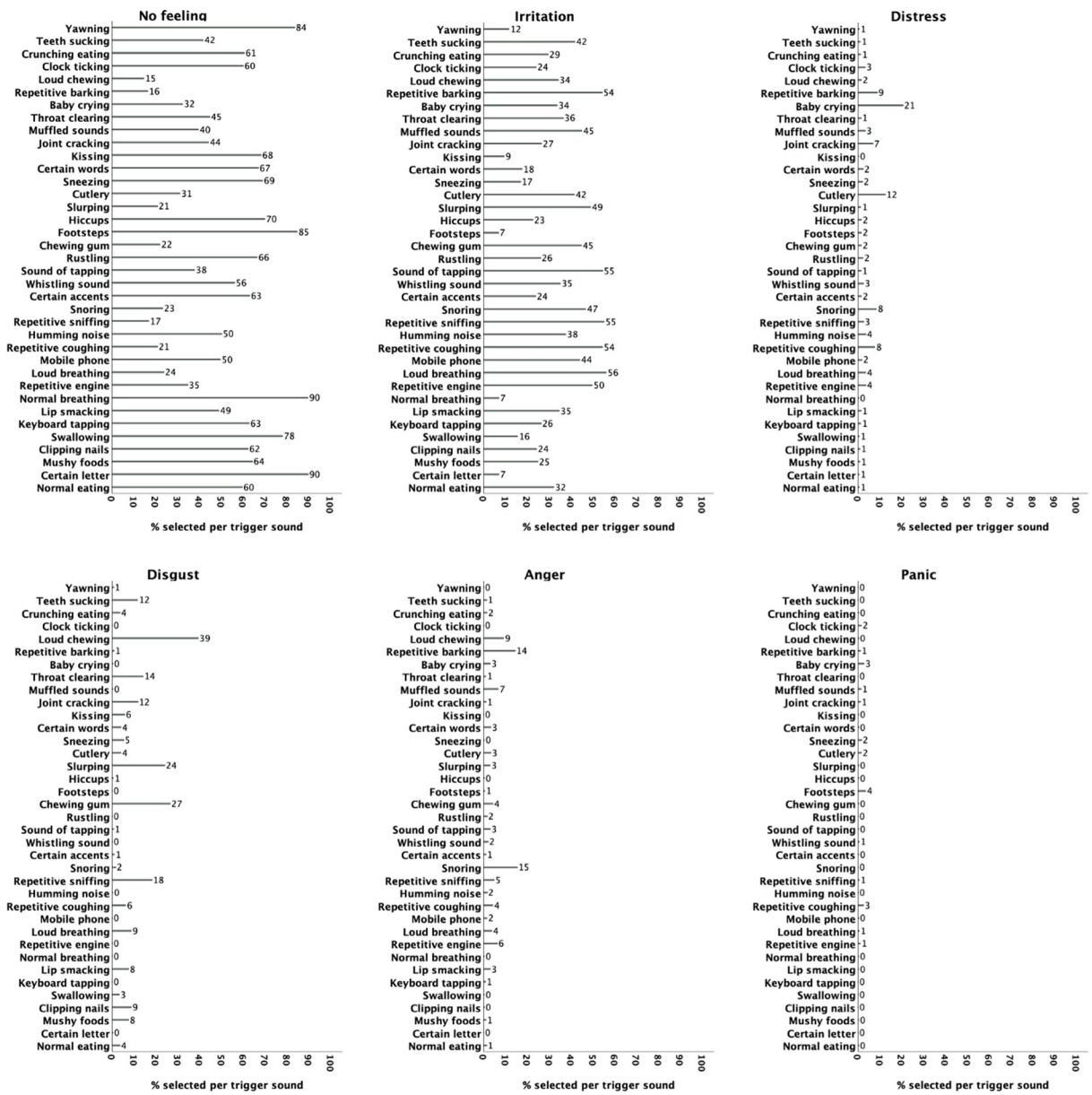

**Fig 2. Percentages of selection per feeling per trigger.**

S-Five factors, A-MISO-S, and MQ, except for no feeling, which showed moderate negative correlations (please see Table 5). Furthermore, the FIRS had moderate to strong correlations with the S-Five factors, A-MISO-R, and MQ. Lower yet significant correlations emerged between the RC scores and the measures of depression and anxiety.

**Intensity of the emotional response.** Table 6 presents the norms and reliability of the intensity items for the 37 S-Five-T sounds. The highest mean intensity occurred in the 'loud chewing', 'repetitive barking', 'snoring', and 'chewing gum' sounds (see also Fig 2). Females reported higher intensity than males in all triggers, some were statistically significant, with the largest differences occurring in 'loud chewing', 'joint cracking', and 'slurping'. Very weak yet significant correlations with age emerged. The stability of the intensity items was excellent (ICC>0.8).

**Table 4. Norms and reliability of the S-Five-T scores.**

| S-Five RC (N = 752) | mean (sd) | median (Q1-Q3) | mode (min-max) | gender difference mean (se)‡ | age rho | Psi (95% CI) | ICC |
|---|---|---|---|---|---|---|---|
| No feeling | 18.7 (6.7) | 18 (14–23) | 17 (1–37) | *-1.4 (0.5) | 0.06 | 0.85 (0.82,1) | 0.87 |
| Irritation | 12 (5.0) | 12 (9–15) | 14 (0–29) | 0.5 (0.4) | **0.19 | 0.82 (0.79,1) | 0.86 |
| Distress | 1.2 (1.6) | 1 (0–2) | 0 (0–9) | **0.3 (0.1) | **-0.17 | 0.79 (0.76,1) | 0.85 |
| Disgust | 2.2 (2.5) | 1 (0–4) | 0 (0–12) | **0.8 (0.2) | **-0.10 | 0.82 (0.80,1) | 0.87 |
| Anger | 1.1 (1.7) | 0 (0–2) | 0 (0–16) | 0 (0.1) | **-0.17 | 0.77 (0.75,1) | 0.85 |
| Panic | 0.3 (0.8) | 0 (0–0) | 0 (0–7) | 0 (0.1) | **-0.25 | 0.69 (0.67,1) | 0.82 |
| TC | 17.5 (6.5) | 18 (13–22) | 17 (0–36) | *1.7 (0.5) | -0.01 | 0.85 (0.83,1) | 0.88 |
| FIRS | 65 (41.6) | 58 (32–88) | 58 (1–241) | *9 (3.1) | **-0.15 | 0.85 (0.82,1) | 0.87 |
| RIRS | 3.6 (1.5) | 3 (2–5) | 2 (0–10) | 0.2 (0.1) | **-0.19 | 0.82 (0.79,1) | 0.86 |

RC: Response count; TC total count; FIRS frequency and intensity reaction count; RIRS relative intensity of reactions score; sd standard deviation; Q1 Q3 first and third quartile; rho: Spearman's correlation coefficient; ICC intraclass correlation coefficient; Psi coefficient and 95% confidence intervals

*p<0.05

**p<0.01

‡ mean difference (se) male vs female comparison, p-value via Mann Whitney test.

Women had higher FIRS scores than men by 9 units on average (Table 6), while a very low and negative correlation with age emerged. FIRS correlated very strongly and negatively with the 'no feeling' RC, indicating convergence validity of the scores. Moderate to strong correlations emerged with other RC scores and had moderate to strong correlations with the S-Five factor and total scores (Table 5). FIRS had low correlations with GAD7 and PHQ9, and moderate with A-MISO-S and moderately strong correlations with the MQ symptom scale.

With respect to the total RIRS score (Table 6), no significant associations with age and gender emerged. RIRS correlated negatively with the 'no feeling' RC, indicating convergence validity of the scores. Moderate to strong correlations emerged with the other RC scores, the S-Five factor and total scores, and the scores of the other measures (Table 5).

## Prevalence of misophonia in the UK population

The Pre-OK-SCIM was implemented with 29 participants sampled from the general population and 26 from the misophonia population (five individuals did not respond to our invitation). The two samples did not significantly differ by age (t = 0.55, df = 53, p = 0.59), with average age of 44 years old (mean = 44.0, sd = 13.7). Of the general sample, 44.8% identified as women, compared to 84.6% women in the misophonia sample ($\chi^2$ (1, N = 55) = 9.3786, p = 0.002). The Pre-OK-SCIM classified 14% (4 participants) of the general population sample and 81% (21 participants) of the misophonia sample as having 'significant misophonia'.

The ROC analysis suggested a cut-off score for the S-Five total as 87 or above (out of 250) for the presence of significant misophonia, where sensitivity and specificity were most balanced (sensitivity 84%, specificity 72%, Youden's J value 0.564, AUC = 0.83). A large AUC was also present for the impact (AUC = 0.87) and threat scores (AUC = 0.88), while the least discriminative of the S-Five scores was the externalising factor (Fig 3). Moderately discriminative were the S-Five-T scores (S1 Fig).

With 87 as the cut-off point on the S-Five, we found that 142 individuals out of 772 met the threshold for significant misophonia in our UK representative sample. Therefore, we estimated that a percentage of 18.4% of the UK population experiences misophonia to an extent that it causes significant burden. There were no significant differences in gender in the prevalence of

**Table 5. Intercorrelations of the S-Five, S-Five-T scores, and correlations with other measures.**

| | No feeling | Irritation | Distress | Disgust | Anger | Panic | TC | FITS | RIRS |
|---|---|---|---|---|---|---|---|---|---|
| **S-Five RC (N = 752)** | | | | | | | | | |
| No feeling | | **-0.61 | **-0.43 | **-0.51 | **-0.44 | **-0.34 | **-0.97 | **-0.76 | **-0.30 |
| Irritation | | | 0.05 | -0.04 | 0.01 | -0.06 | **0.67 | **0.33 | *-0.09 |
| Distress | | | | **0.26 | **0.16 | **0.25 | **0.41 | **0.38 | **0.21 |
| Disgust | | | | | **0.39 | **0.23 | **0.50 | **0.59 | **0.42 |
| Anger | | | | | | **0.28 | **0.43 | **0.57 | **0.47 |
| Panic | | | | | | | **0.29 | **0.35 | **0.28 |
| TC | | | | | | | | **0.76 | **0.28 |
| FITS | | | | | | | | | **0.81 |
| **S-Five Factors (N = 752)** | | | | | | | | | |
| Externalising | **-0.35 | **0.11 | **0.13 | **0.35 | **0.37 | **0.16 | **0.36 | **0.48 | **0.41 |
| Internalising | **-0.36 | **0.10 | **0.26 | **0.28 | **0.33 | **0.26 | **0.37 | **0.50 | **0.43 |
| Impact | **-0.31 | *0.09 | **0.26 | **0.22 | **0.26 | **0.26 | **0.31 | **0.43 | **0.38 |
| Outburst | **-0.35 | 0.07 | **0.25 | **0.29 | **0.40 | **0.25 | **0.35 | **0.48 | **0.41 |
| Threat | **-0.40 | *0.08 | **0.36 | **0.31 | **0.37 | **0.30 | **0.39 | **0.51 | **0.43 |
| Total | **-0.44 | **0.10 | **0.30 | **0.38 | **0.44 | **0.30 | **0.44 | **0.60 | **0.52 |
| **A-MISO-S (N = 319)** | | | | | | | | | |
| Total | **-0.36 | *0.13 | **0.18 | **0.32 | **0.37 | **0.21 | **0.38 | **0.44 | **0.32 |
| **MQ (N = 300)** | | | | | | | | | |
| MSYS (N = 261) | **-0.52 | **0.27 | **0.18 | **0.38 | **0.37 | **0.19 | **0.54 | **0.56 | **0.33 |
| MEBS (N = 261) | **-0.39 | 0.05 | **0.19 | **0.27 | **0.44 | **0.33 | **0.39 | **0.49 | **0.43 |
| MSES (N = 300) | **-0.33 | 0.08 | **0.18 | **0.23 | **0.31 | **0.27 | **0.31 | **0.38 | **0.29 |
| Total (N = 261) | **-0.52 | **0.16 | **0.21 | **0.37 | **0.47 | **0.31 | **0.51 | **0.60 | **0.46 |
| **PHQ9 (N = 726)** | | | | | | | | | |
| Total | **-0.29 | 0.07 | **0.25 | **0.22 | **0.25 | **0.24 | **0.28 | **0.31 | **0.22 |
| **GAD7 (N = 736)** | | | | | | | | | |
| Total | **-0.22 | 0.02 | **0.22 | **0.23 | **0.20 | **0.23 | **0.22 | **0.26 | **0.19 |

RC: Response count; TC total count; FIRS frequency and intensity reaction count; RIRS relative intensity of reactions score; rho: Spearman's correlation coefficient; A-MISO-S: Amsterdam Misophonia Scale; MQ: Misophonia Questionnaire; MSYS: Misophonia Symptoms Scale; MEBS: Misophonia Emotions and Behaviours Scale; MSES: Misophonia Severity Scale; PHQ-9: Physical Health Questionnaire; GAD-7: Generalised Anxiety Disorder Assessment.

misophonia ($\chi^2$ (1, N = 768) = 0.06, p = 0.80). The average age of those above the threshold for misophonia (mean = 43.7 years; SD = 1.21) was lower than those below the threshold for misophonia (mean = 47.0 years; SD = 0.63), and this difference was significant (t = 2.18, df = 770, p = 0.03). The age and gender of participants were not found to significantly affect either the performance of the S-Five total score or the ability of the S-Five total score to discriminate between cases and controls.

## Discussion

The main purpose of this study was to evaluate the structure and psychometric properties of the scale, present the S-Five and S-Five-T norms for the general UK population, and to estimate the prevalence of misophonia in the UK.

Factor analyses supported the five-factor structure originally validated in a sample of individuals identifying with misophonia [22], suggesting that this tool is suitable for use in both clinical and community samples. Measurement invariance with respect to age and gender was established. The scale showed satisfactory reliability indices (internal consistency and stability)

**Table 6. Norms and reliability of the intensity items for the 37 S-Five-T sounds.**

| Trigger sounds | mean (sd) | median (Q1-Q3) | mode (min-max) | Average gender difference‡ | age rho | Psi (95% CI) | ICC |
|---|---|---|---|---|---|---|---|
| Normal eating sounds | 0.90 (1.8) | 0 (0–1) | 0 (0–10) | -0.2 (0.1) | -0.1 | 0.74 (0.72,1) | 0.84 |
| Certain letter sounds | 0.30 (1.1) | 0 (0–0) | 0 (0–10) | -0.1 (0.1) | -0.1 | 0.69 (0.67,1) | 0.82 |
| Mushy foods | 1.27 (2.2) | 0 (0–2) | 0 (0–10) | -0.1 (0.2) | **-0.2 | 0.74 (0.71,1) | 0.84 |
| Sound of clipping nails | 1.16 (2.0) | 0 (0–2) | 0 (0–10) | *-0.6 (0.1) | 0 | 0.77 (0.75,1) | 0.85 |
| Swallowing | 0.71 (1.7) | 0 (0–0) | 0 (0–10) | -0.2 (0.1) | **-0.2 | 0.73 (0.71,1) | 0.83 |
| Keyboard tapping | 0.88 (1.7) | 0 (0–1) | 0 (0–10) | -0.1 (0.1) | 0 | 0.74 (0.72,1) | 0.84 |
| Lip smacking | 1.66 (2.3) | 0 (0–3) | 0 (0–10) | **-0.5 (0.2) | **-0.2 | 0.78 (0.75,1) | 0.85 |
| Normal breathing | 0.23 (1.0) | 0 (0–0) | 0 (0–9) | 0 (0.1) | **-0.1 | 0.67 (0.67,1) | 0.82 |
| Repetitive engine | 2.14 (2.3) | 2 (0–3) | 0 (0–10) | 0 (0.2) | 0 | 0.79 (0.76,1) | 0.85 |
| Blocked nose | 2.67 (2.4) | 2 (0–4) | 0 (0–10) | **-0.5 (0.2) | **-0.2 | 0.80 (0.77,1) | 0.86 |
| Mobile phone | 1.48 (2.0) | 0 (0–3) | 0 (0–10) | 0 (0.2) | *0.1 | 0.76 (0.74,1) | 0.84 |
| Repetitive coughing | 2.84 (2.4) | 2 (1–5) | 0 (0–10) | -0.1 (0.2) | 0 | 0.79 (0.77,1) | 0.85 |
| Humming | 1.46 (2.1) | 0 (0–2) | 0 (0–9) | -0.2 (0.2) | 0 | 0.77 (0.75,1) | 0.85 |
| Repetitive sniffing | 3.28 (2.6) | 3 (1–5) | 0 (0–10) | **-0.6 (0.2) | 0 | 0.83 (0.80,1) | 0.87 |
| Snoring | 3.54 (3.0) | 3 (0–6) | 0 (0–10) | **-0.9 (0.2) | 0 | 0.83 (0.81,1) | 0.87 |
| Certain accents | 0.97 (1.9) | 0 (0–1) | 0 (0–10) | *0.3 (0.1) | 0.1 | 0.73 (0.70,1) | 0.83 |
| Whistling sound | 1.30 (2.1) | 0 (0–2) | 0 (0–10) | -0.2 (0.2) | 0 | 0.76 (0.74,1) | 0.84 |
| Tapping | 1.99 (2.4) | 1 (0–3) | 0 (0–10) | *-0.4 (0.2) | 0 | 0.81 (0.79,1) | 0.86 |
| Rustling plastic or paper | 0.99 (1.9) | 0 (0–1) | 0 (0–10) | 0 (0.1) | -0.1 | 0.76 (0.74,1) | 0.84 |
| Chewing gum | 3.31 (2.8) | 3 (1–5) | 0 (0–10) | **-0.5 (0.2) | 0 | 0.79 (0.77,1) | 0.85 |
| Footsteps | 0.50 (1.5) | 0 (0–0) | 0 (0–10) | 0.1 (0.1) | **-0.2 | 0.71 (0.69,1) | 0.83 |
| Hiccups | 0.67 (1.5) | 0 (0–1) | 0 (0–9) | 0.1 (0.1) | 0 | 0.75 (0.72,1) | 0.84 |
| Slurping | 2.99 (2.6) | 3 (1–5) | 0 (0–10) | **-0.8 (0.2) | 0 | 0.80 (0.78,1) | 0.86 |
| Cutlery | 2.87 (2.9) | 2 (0–5) | 0 (0–10) | **-0.6 (0.2) | **-0.2 | 0.81 (0.78,1) | 0.86 |
| Sneezing | 0.99 (1.9) | 0 (0–1) | 0 (0–10) | 0.1 (0.1) | 0 | 0.74 (0.72,1) | 0.84 |
| Certain words | 1.13 (2.1) | 0 (0–2) | 0 (0–10) | 0 (0.2) | *0.1 | 0.75 (0.73,1) | 0.84 |
| Kissing | 0.58 (1.6) | 0 (0–0) | 0 (0–10) | **-0.4 (0.1) | **-0.2 | 0.72 (0.7,1) | 0.83 |
| Joint cracking | 1.83 (2.5) | 0 (0–3) | 0 (0–10) | **-0.9 (0.2) | *0.1 | 0.80 (0.77,1) | 0.86 |
| Muffled sounds | 2.16 (2.5) | 2 (0–3) | 0 (0–10) | 0.2 (0.2) | 0 | 0.79 (0.76,1) | 0.85 |
| Throat clearing | 1.89 (2.4) | 1 (0–3) | 0 (0–10) | *-0.4 (0.2) | 0 | 0.78 (0.75,1) | 0.85 |
| Baby crying | 2.78 (2.8) | 2 (0–5) | 0 (0–10) | 0 (0.2) | **-0.2 | 0.83 (0.80,1) | 0.87 |
| Repetitive barking | 3.63 (2.8) | 3 (1–6) | 0 (0–10) | -0.2 (0.2) | *0.1 | 0.81 (0.79,1) | 0.86 |
| Loud chewing | 4.21 (3.0) | 4 (2–6) | 0 (0–10) | **-0.8 (0.2) | *-0.1 | 0.84 (0.82,1) | 0.87 |
| Clock ticking | 1.02 (2.0) | 0 (0–1) | 0 (0–10) | -0.1 (0.2) | *-0.1 | 0.77 (0.74,1) | 0.84 |
| Crunching | 1.29 (2.2) | 0 (0–2) | 0 (0–10) | -0.2 (0.2) | *-0.1 | 0.78 (0.75,1) | 0.85 |
| Teeth sucking | 2.06 (2.5) | 1 (0–3) | 0 (0–10) | **-0.5 (0.2) | 0 | 0.80 (0.78,1) | 0.86 |
| Yawning | 0.44 (1.3) | 0 (0–0) | 0 (0–9) | -0.1 (0.1) | 0 | 0.72 (0.7,1) | 0.83 |

sd standard deviation; Q1 Q3 first and third quartile; rho: Spearman's correlation coefficient; ICC intraclass correlation coefficient; Psi coefficient and 95% confidence intervals

*p<0.05

**p<0.01; ‡ mean difference (se) male vs female comparison, p-value via Mann Whitney test.

and concurrent (convergent and discriminant) validity. In all assessments, the S-Five was found to have satisfactory psychometric properties for the UK general population.

Average scores were highest for the externalising appraisals, and lowest in the impact factor. This is in contrast to the pattern of findings found in the population of individuals identifying

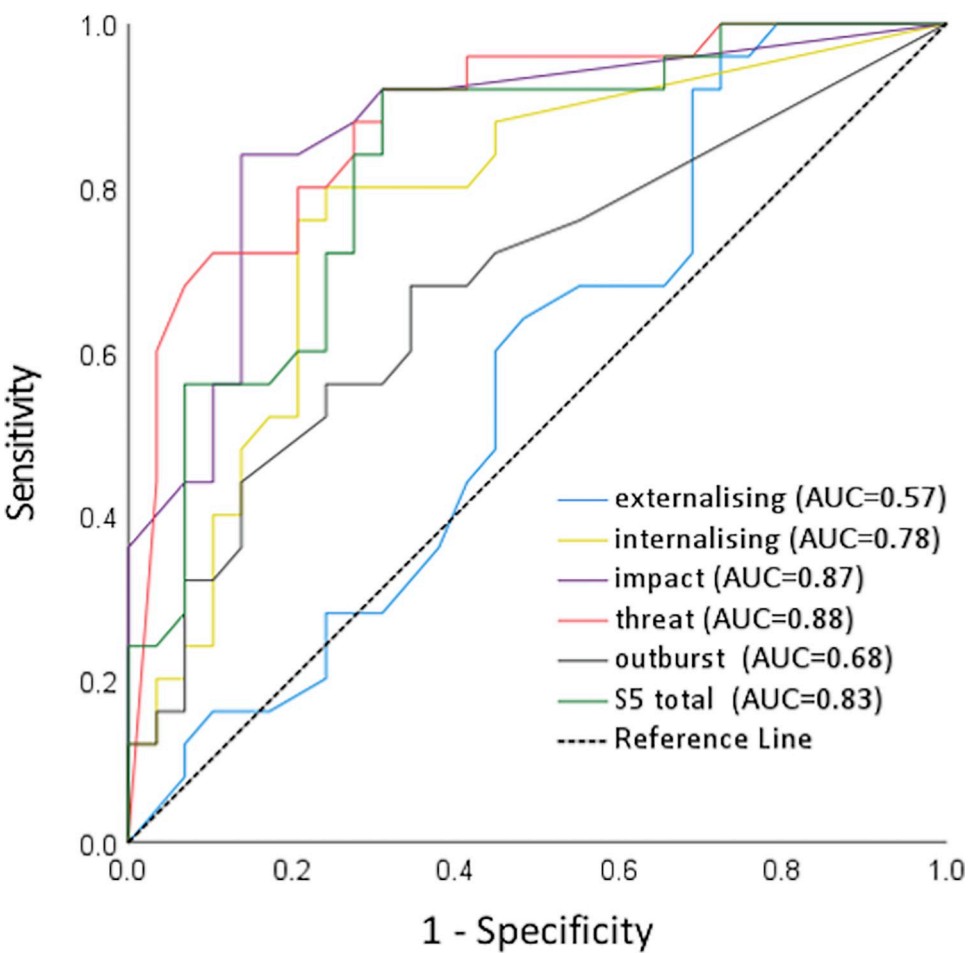

**Fig 3. ROC curves and the estimated area under the curve (AUC) for each of the S-Five scores.**

with the condition [22], where the mean scores were the highest for the threat factor while the lowest for the outburst factor. These results indicate that attributing blame on others for making aversive sounds is a relatively common phenomenon, compared with the other factors measured by the S-Five. This is supported by the finding that the externalising factor showed to be the least discriminative of misophonia.

In relation to trigger sounds, we found that there were certain sounds that frequently elicited a negative emotional response, such as loud chewing, slurping, snoring and loud breathing. The most common reaction reported was irritation, except in the case of loud chewing, where disgust was more frequent. This suggests that many of the sounds frequently reported as triggers in misophonia are also aversive to the general population. However, there appear to be two key differences in the pattern of trigger reactions in misophonia compared with the general population. The first is in the types of triggers, for example, normal breathing being indicative of higher levels of misophonia, and swallowing being a highly reliable indicator for individuals with higher misophonia levels [24]. Both these sounds were reported as eliciting no feeling in most of the general population in the present study.

The second key difference is in the nature of the reaction, with anger and panic reported more often in misophonia [22]. We found that irritation was frequently reported by the general population for a range of sounds, and that the irritation reaction count had only a low

correlation with the total S-Five and the A-MISO-S, providing further evidence that respond-ing with irritation is not a good indicator of the presence of misophonia [22,55]. Additionally, irritation was not correlated with any of the other negative emotion reaction counts (anger, disgust, distress and panic), nor with symptoms of anxiety and depression, all of which were moderately correlated with each other and with S-five total scores. This study estimated that almost one in five people (18.4%) in the UK experience misophonia to a level where it causes significant burden in terms of distress and impact. The prevalence and severity of misophonia appears to be similar in men and women, which was surprising considering the disproportion-ate number of women in previous misophonia research samples [see for instance 3,18,56]. Fur-ther, there was a small but significant difference in the average age for those with significant and non-significant misophonia with those above the threshold for misophonia being on aver-age 3.3 years younger than those below the threshold. This is consistent with the prevalence study in Ankara, Turkey [20], which indicated younger age to be a predictor of misophonia. This could be due to improved coping strategies and developing more effective organisation of everyday functioning to minimise the exposure to sounds and their impact as the age progresses.

It is important to note that the preliminary version of the OK-SCIM was developed prior to the consensus definition of misophonia as a disorder [1]. Our research team have refined the OK-SCIM and are validating the tool's capacity to distinguish clinical misophonia (i.e. miso-phonia that causes current significant distress or impairment at a "disordered" level) from sub-clinical misophonia (presence of misophonia symptoms without current significant distress and impairment) and no misophonia. This will enable us to estimate prevalence of misophonia as a disorder as well as significant symptoms of misophonia. The present study should there-fore be interpreted as the prevalence of individuals who have misophonia symptoms to an extent that they consider it to cause a significant burden in their lives.

The strength of the present study is that a large sample representative of the UK general population was used, which contributed to the external validity of the results. Furthermore, state-of-the-art psychometric techniques were used, including ROC curve regression analysis, which allowed us to establish a meaningful cut-off score for significant misophonia, found not to be affected by age or gender. We therefore present the best estimate available for misopho-nia in the UK.

There were several limitations to this study. First, the sample was representative of the UK population only; the results may differ across countries and cultures. Second, there were limi-tations to the interview protocol, as described above. We attempted to minimise the limitations of this tool by having it administered by qualified psychologists experienced in working with misophonia, who were able to use the tool flexibly to determine the presence or absence of sig-nificant misophonia. The OK-SCIM has since been refined by the research team in line with the consensus definition of misophonia as a disorder [1], and subsequently tested for use by non-clinician interviewers, to validate its capacity to identify subclinical and clinical mispho-nia (data currently being analysed). Finally, the questionnaire should be tested in a treatment sample to assess its suitability for use as a measure of clinical change.

In conclusion, our results show that the S-Five is a valid and reliable tool for measuring the presence and severity of misophonia in the UK general population. The Vitoratou [22] five-factor solution was replicated in the general population data, with good reliability and validity. By using semi-structured clinical interviews, we were able to establish a cut-off score for signif-icantly burdensome misophonia. This, in turn, allowed us to estimate that the prevalence of misophonia in the UK is 18.4%. Our results show that misophonia is relatively common condi-tion and further research is needed to determine at what point this condition becomes "disor-dered" in terms of distress, impact and need for treatment.

## Supporting information

**S1 Fig. ROC curves and the estimated area under the curve (AUC) for each of the S-Five-T scores.**
(TIF)

**S1 Table. The S-Five statements.**
(DOCX)

**S2 Table. Scoring instructions for the S-Five.**
(DOCX)

**S3 Table. The S-Five-t triggers checklist and scoring.**
(DOCX)

**S4 Table. Full list of administered questionnaires.**
(DOCX)

## Author Contributions

**Conceptualization:** Silia Vitoratou, Jane Gregory.

**Data curation:** Silia Vitoratou.

**Formal analysis:** Silia Vitoratou, Chloe Hayes.

**Funding acquisition:** Silia Vitoratou.

**Investigation:** Silia Vitoratou, Chloe Hayes, Tom Graham, Jane Gregory.

**Methodology:** Silia Vitoratou, Chloe Hayes, Nora Uglik-Marucha, Tom Graham, Jane Gregory.

**Project administration:** Jane Gregory.

**Software:** Silia Vitoratou.

**Supervision:** Silia Vitoratou, Jane Gregory.

**Validation:** Silia Vitoratou.

**Visualization:** Silia Vitoratou.

**Writing – original draft:** Silia Vitoratou, Chloe Hayes, Nora Uglik-Marucha, Oliver Pearson, Jane Gregory.

**Writing – review & editing:** Silia Vitoratou, Chloe Hayes, Nora Uglik-Marucha, Oliver Pearson, Tom Graham, Jane Gregory.

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
