## [Decision Letter · Decision Letter 0]

17 Oct 2022

PONE-D-22-17376Misophonia in the UK: Prevalence and norms from the S-Five in a UK representative samplePLOS ONE

Dear Dr. Vitoratou,

Thank you for submitting your manuscript to PLOS ONE. After careful consideration, we feel that it has merit but does not fully meet PLOS ONE’s publication criteria as it currently stands. Therefore, we invite you to submit a revised version of the manuscript that addresses the points raised during the review process. First of all, apologies for the delay in the feedback. It has been considerably challenging to locate a suitable (and available) reviewer for this submission. Your paper has been now reviewed by an acknowledged expert, who provides a set of interesting comments, queries, and suggestions. Please carefully address and respond to all of them in both the rebuttal letter and the revised version of the manuscript, making the changes visible through a suitable tracking method.

We look forward to receiving your revised manuscript.

Kind regards,

Sergio A. Useche, Ph.D.

Academic Editor

PLOS ONE

Journal Requirements:

a) Did participants provide their written or verbal informed consent to participate in this study?

SV, CH, OP and NUM were funded or partially funded by the Biomedical Research Centre for Mental Health at South London and Maudsley NHS Foundation Trust and King’s College London.  This research was funded in whole, or in part, by the Wellcome Trust [JG; Grant number 102176/B/13/Z]. For the purpose of open access, the author has applied a CC BY public copyright licence to any Author Accepted Manuscript version arising from this submission. The views expressed are those of the author(s) and not necessarily those of the NHS, The Wellcome Trust, the NIHR or the Department of Health and Social Care.

SV, CH, OP and NUM were funded or partially funded by the Biomedical Research Centre for Mental Health at South London and Maudsley NHS Foundation Trust and King’s College London.  This research was funded in whole, or in part, by the Wellcome Trust [JG; Grant number 102176/B/13/Z]. For the purpose of open access, the author has applied a CC BY public copyright licence to any Author Accepted Manuscript version arising from this submission. The views expressed are those of the author(s) and not necessarily those of the NHS, The Wellcome Trust, the NIHR or the Department of Health and Social Care.

The authors declare that the research was conducted in the absence of any commercial or financial relationships that could be construed as a potential conflict of interest. 

6. We noted in your submission details that a portion of your manuscript may have been presented or published elsewhere. Please clarify whether this  publication was peer-reviewed and formally published. If this work was previously peer-reviewed and published, in the cover letter please provide the reason that this work does not constitute dual publication and should be included in the current manuscript.

7. We note that you have indicated that data from this study are available upon request. PLOS only allows data to be available upon request if there are legal or ethical restrictions on sharing data publicly. For more information on unacceptable data access restrictions, please see http://journals.plos.org/plosone/s/data-availability#loc-unacceptable-data-access-restrictions. 

8. We note that you have referenced (ie. Bewick et al. [5]) which has currently not yet been accepted for publication. Please remove this from your References and amend this to state in the body of your manuscript: (ie “Bewick et al. [Unpublished]”) as detailed online in our guide for authors

Reviewers' comments:

Reviewer's Responses to Questions

**Comments to the Author**

1. Is the manuscript technically sound, and do the data support the conclusions?

Reviewer #1: Yes

2. Has the statistical analysis been performed appropriately and rigorously? 

Reviewer #1: Yes

3. Have the authors made all data underlying the findings in their manuscript fully available?

Reviewer #1: Yes

4. Is the manuscript presented in an intelligible fashion and written in standard English?

Reviewer #1: Yes

5. Review Comments to the Author

Reviewer #1: The current manuscript represents a psychometric evaluation of the S-Five in a UK sample, with a goal of developing a clinically significant symptoms threshold that is consistent with clinician consensus. A secondary goal is to use this threshold to measure the UK prevalence of misophonia. While the study itself is straightforward, the introductory framing leaves a bit to be desired, and some care should be taken to the formatting and presentation of results to make them more interpretable to a reader not intimately familiar with each of the S-Five items.

Introduction

Page 3 line 69 the authors refer to their previous work in the third person and then refer to their current work in the next sentence in the first person. It is clear this is the same group of people (most authors overlap) so it would make sense to say something like “our research team used the responses…” and “our work resulted” to make it clear this is a continuation of an ongoing body of work rather than building on someone’s else’s work.

Additional justification for why a prevalence rate is needed specifically for UK population would be helpful. Although prevalence has varied across studies on misophonia, most report somewhere between 12-20%, similar to the studies cited here. Do the authors expect the UK to be different? The more interesting question here is how the S-Five corresponds to semi-structured interviews and clinician consensus to determine a severity or diagnostic threshold for clinically significant symptoms, yet the majority of the introduction is focused on prevalence. Given that the S-Five, while very useful for addressing intensity of a misophonic response, is currently limited compared to scales like the MQ because it does not have a diagnostic threshold score, this study represents a major step forward in usability of the scale. I would suggest refocusing the introduction a bit to highlight the psychometric aspects of this study rather than just getting a prevalence rate.

Methods

Methods are appropriate as described, however I am curious as to why the authors omitted the Anxiety Sensitivity Index from their targeted analyses, since it was administered according to the Supplement, and anxiety is highly co-occuring in the misophonia population. Might it not be more sensitive than the GAD-7, particularly when it comes to anticipatory anxiety, which is also more common in misophonia than the general population?

Results

The sample skews quite old compared to some of the other samples (most of the larger studies were in undergraduate students). Can the authors comment on whether this was expected to impact responses or prevalence ratings and whether this is comparable to the other studies?

In Table 1 the ICC for every single question is 0.8. Is that really true or a typo?

Table 5 some of the words in the top header bar are cut off.

Table 6 the fourth item is just listed as “sound of”? Several of the items are listed as “normal” or “repetitive” which makes it difficult to know what they are referring to from the table alone. Please modify this column to be more descriptive.

Discussion

The authors discuss frequency of reporting negative emotional response and its variation by sound type, with some sounds being more frequently reported in the general population as at least irritating, and some sounds, mostly mouth sounds being more discriminative of misophonia. These findings are largely consistent with a recent study in a US sample that used the older S-Five (so not perfectly comparable, but the trigger sounds of greatest importance are the same). This may help the author’s case that this finding is a general one and remove the limitation that this particular finding at least might be specific to the UK population. (Norris JE, Kimball SH, Nemri DC and Ethridge LE (2022) Toward a Multidimensional Understanding of Misophonia Using Cluster-Based Phenotyping. Front. Neurosci. 16:832516. doi: 10.3389/fnins.2022.832516).

The results regarding depression and anxiety questionnaires and their correspondence to S-Five scores is not discussed. They deserve at least a short mention.

6. PLOS authors have the option to publish the peer review history of their article (what does this mean?). If published, this will include your full peer review and any attached files.

Reviewer #1: No

---

## [Author Response · Author response to Decision Letter 0]

27 Jan 2023

Dear Editor,

We greatly appreciate your and reviewers’ time to review our manuscript and offer your valuable suggestions. We have carefully gone through each of your comments and addressed them to the best of our ability resulting in modifications to the manuscript according to your feedback. This has led to significant improvements of the paper for which we are thankful. 

Please find below point-by-point comments addressed:

Reviewer 1

Reviewer #1: The current manuscript represents a psychometric evaluation of the S-Five in a UK sample, with a goal of developing a clinically significant symptoms threshold that is consistent with clinician consensus. A secondary goal is to use this threshold to measure the UK prevalence of misophonia. While the study itself is straightforward, the introductory framing leaves a bit to be desired, and some care should be taken to the formatting and presentation of results to make them more interpretable to a reader not intimately familiar with each of the S-Five items.

Introduction

1. Page 3 line 69 the authors refer to their previous work in the third person and then refer to their current work in the next sentence in the first person. It is clear this is the same group of people (most authors overlap) so it would make sense to say something like “our research team used the responses…” and “our work resulted” to make it clear this is a continuation of an ongoing body of work rather than building on someone’s else’s work.

 Corrected: we followed the reviewer’s advice 

2. Additional justification for why a prevalence rate is needed specifically for UK population would be helpful. Although prevalence has varied across studies on misophonia, most report somewhere between 12-20%, similar to the studies cited here. Do the authors expect the UK to be different? 

Response: As our team is based in the UK, we started studying misophonia in the UK. However, our team is currently initiating research to estimate the prevalence of misophonia in the US. Also, we are currently translating the S-Five in several languages and aim to estimate the prevalence of misophonia for other populations as well. There were no a-priori expectations about difference in prevalence of the disorders cross-culturally, but this is a question that interests our team greatly as well and subject of our future research. 

3. The more interesting question here is how the S-Five corresponds to semi-structured interviews and clinician consensus to determine a severity or diagnostic threshold for clinically significant symptoms, yet the majority of the introduction is focused on prevalence. Given that the S-Five, while very useful for addressing intensity of a misophonic response, is currently limited compared to scales like the MQ because it does not have a diagnostic threshold score, this study represents a major step forward in usability of the scale. I would suggest refocusing the introduction a bit to highlight the psychometric aspects of this study rather than just getting a prevalence rate.

Response: Thank you for this suggestion, the introduction has been altered so that the differences which have arisen in previous research regarding defining such a prevalence rate for misophonia are linked to the present study of defining a threshold for clinical misophonia, and thus suggesting a prevalence rate from this. We have revised the manuscript to read (Page 3; Lines 74-81): 

“using a self-report tool, the A-MISO-S [4], which has not been validated as a self-report screening tool. The high prevalence estimated in this sample of medical students, compared with the prevalence found by household interview in Ankara, raises questions about the use of a measure that has not been validated as a screening tool to estimate prevalence, and unclear definitions of clinically significant misophonia. It is also possible that misophonia prevalence is higher in medical students than in the general population, although the discrepancy still highlights the importance of validating cut-off scores.”

Methods

4. Methods are appropriate as described, however I am curious as to why the authors omitted the Anxiety Sensitivity Index from their targeted analyses, since it was administered according to the Supplement, and anxiety is highly co-occurring in the misophonia population. Might it not be more sensitive than the GAD-7, particularly when it comes to anticipatory anxiety, which is also more common in misophonia than the general population?

Response: For the present study, we were interested in the association with general symptoms of anxiety (as measured by the GAD7) rather than the trait of anxiety sensitivity, as measured by the ASI-3, which captures relatively stable beliefs about anxiety symptoms and is not a measure of current anxiety symptoms. We selected the GAD7 specifically because this is the measure of general symptoms of anxiety that is used as a routine weekly outcome measure in primary care psychology in the UK, enabling us to view these results in the context of UK mental health care.

The ASI-3 was indeed collected, this was done as part of a Masters’ student’s hypothesis testing about the trait of anxiety sensitivity in relation to specific aspects of misophonia (which has been published elsewhere).

Results

5. The sample skews quite old compared to some of the other samples (most of the larger studies were in undergraduate students). Can the authors comment on whether this was expected to impact responses or prevalence ratings and whether this is comparable to the other studies?

Response: This is a sample representative of the general population in the UK and therefore the only sample that can be used for a general population prevalence, whereas the other samples were able to generalise only to a specific group in terms of age and other characteristics. Subgroups might have their own prevalence rates and this remains to be seen if in future research. We added in the manuscript (Page 23; Lines 515-521) : 

“increased control over exposure to sounds over time. This finding also suggests that previous studies using university student samples, may show higher prevalence than the general population due to the lower age of the samples. The higher prevalence found in the present study than in the Ankara study (ref turkey) also highlights the importance of developing consensus agreement on the clinical threshold for the condition, especially with the recent definition of misophonia as a disorder [1].” 

6. In Table 1 the ICC for every single question is 0.8. Is that really true or a typo?

Response: The ICC ranges from 0.83 to 0.87, with each rounded to 2 decimal places. Perhaps there was an error in how the table was printed in the pdf file and presented to the reviewer. We tried to correct how the tables print in the document and we hope the ICC values are now properly displayed.

7. Table 5 some of the words in the top header bar are cut off.

Corrected: We thank the reviewer for pointing this out. The table has been adjusted so that all headers are printed in clear and readable. 

8. Table 6 the fourth item is just listed as “sound of”? Several of the items are listed as “normal” or “repetitive” which makes it difficult to know what they are referring to from the table alone. Please modify this column to be more descriptive.

Corrected: This column has been reformatted to solve the issue of unclear trigger item labels being printed. 

Discussion

9. The authors discuss frequency of reporting negative emotional response and its variation by sound type, with some sounds being more frequently reported in the general population as at least irritating, and some sounds, mostly mouth sounds being more discriminative of misophonia. These findings are largely consistent with a recent study in a US sample that used the older S-Five (so not perfectly comparable, but the trigger sounds of greatest importance are the same). This may help the author’s case that this finding is a general one and remove the limitation that this particular finding at least might be specific to the UK population. (Norris JE, Kimball SH, Nemri DC and Ethridge LE (2022) Toward a Multidimensional Understanding of Misophonia Using Cluster-Based Phenotyping. Front. Neurosci. 16:832516. doi: 10.3389/fnins.2022.832516).

We thank the reviewer for bringing to our attention this new research article, which we have now incorporated in the manuscript. Page 24 (lines 506-509) of the manuscript now reads: 

“However, recent evidence reporting a similar response pattern to trigger sounds has been found in a US sample [57], which also used the S-Five to assess misophonia, supporting the ability to generalise the results of this study to other countries. This remains an important focus for future research to investigate.”

10. The results regarding depression and anxiety questionnaires and their correspondence to S-Five scores is not discussed. They deserve at least a short mention.

Thank you for highlighting that we did not discuss the results of the depression and anxiety questionnaires. We have now included this in the manuscript (Page 21; lines 476-483), which reads: 

“All factors were positively correlated with depression and anxiety, with both being moderately correlated with the threat factor, and low correlations with all other factors. This supports previous research finding an association between misophonia and anxiety and depression ref – see Potgieter et al for review), and also provides preliminary evidence that this may relate to the threat factor, in particular.”

In addition to the reviewers’ comments, we have modified the manuscript to PLOS ONE journal requirements, which has been highlighted during the revision process. 

The manuscript has been modified according to PLOS ONE’s style requirements. Specifically, the corresponding author’s initials have been added in parentheses after the e-mail address (page 1, line 27).

a) Did participants provide their written or verbal informed consent to participate in this study?

Thank you for this reminder. We have now updated the manuscript to reflect this, and it reads as follows (Page 4, lines 97-101):

Using an online survey platform, participants read and provided written consent to the participants’ information sheet (ethics approval reference RESCM-19/20-11826) and were subsequently screened for their eligibility criteria, which included being aged 18 years or older, English fluency, and no diagnosis of a severe learning or intellectual disability.

SV, CH, OP and NUM were funded by the Biomedical Research Centre for Mental Health at South London and Maudsley NHS Foundation Trust and King’s College London. This research was funded in whole, or in part, by the Wellcome Trust [JG; Grant number 102176/B/13/Z]. For the purpose of open access, the author has applied a CC BY public copyright licence to any Author Accepted Manuscript version arising from this submission. The views expressed are those of the author(s) and not necessarily those of the NHS, The Wellcome Trust, the NIHR or the Department of Health and Social Care.

SV, CH, OP and NUM were funded or partially funded by the Biomedical Research Centre for Mental Health at South London and Maudsley NHS Foundation Trust and King’s College London. This research was funded in whole, or in part, by the Wellcome Trust [JG; Grant number 102176/B/13/Z]. For the purpose of open access, the author has applied a CC BY public copyright licence to any Author Accepted Manuscript version arising from this submission. The views expressed are those of the author(s) and not necessarily those of the NHS, The Wellcome Trust, the NIHR or the Department of Health and Social Care.

Please see the amended Funding Statement below:

“SV, CH, OP and NUM were funded or partially funded by the Biomedical Research Centre for Mental Health at South London and Maudsley NHS Foundation Trust and King’s College London. This research was funded in whole, or in part, by the Wellcome Trust [JG; Grant number 102176/B/13/Z]. For the purpose of open access, the author has applied a CC BY public copyright licence to any Author Accepted Manuscript version arising from this submission. The views expressed are those of the author(s) and not necessarily those of the NHS, The Wellcome Trust, the NIHR or the Department of Health and Social Care. The funders had no role in study design, data collection and analysis, decision to publish, or preparation of the manuscript. There was no additional external funding received for this study.”

The authors declare that the research was conducted in the absence of any commercial or financial relationships that could be construed as a potential conflict of interest. 

With respect to Competing Interests section, we would like to state that “the authors have declared that no competing interests exist”.

6. We noted in your submission details that a portion of your manuscript may have been presented or published elsewhere. Please clarify whether this publication was peer-reviewed and formally published. If this work was previously peer-reviewed and published, in the cover letter please provide the reason that this work does not constitute dual publication and should be included in the current manuscript.

The manuscript has been submitted as a pre-print to PsyArXiv (doi: 10.31234/osf.io/jwpeq) but has not been peer-reviewed or formally published in any journal, thus it does not constitute dual publication.

7. We note that you have indicated that data from this study are available upon request. PLOS only allows data to be available upon request if there are legal or ethical restrictions on sharing data publicly. For more information on unacceptable data access restrictions, please see http://journals.plos.org/plosone/s/data-availability#loc-unacceptable-data-access-restrictions. 

Thank you, the data has been anonymised and will be available with the manuscript. The data has been uploaded to a data repository and is awaiting approval. The full reference, with the URL, will be provided as soon as this is accepted. 

Vitoratou, Silia (2022). Misophonia in the UK: Prevalence and norms for the S-Five in a UK representative sample. [Data Collection]. Colchester, Essex: UK Data Service.

8. We note that you have referenced (ie. Bewick et al. [5]) which has currently not yet been accepted for publication. Please remove this from your References and amend this to state in the body of your manuscript: (ie “Bewick et al. [Unpublished]”) as detailed online in our guide for authors

Thank you for highlighting this. The reference has now been removed from the manuscript. 

At the end of the manuscript (Page 31, lines 661-667), we have listed captions from the Supporting Information file. We have also modified the manuscript for in-text citations to Tables in Supporting Information file per the guidelines, which read now as follows:

Page 5, lines 107-109: An extended battery of 17 scales were considered within the S-Five study, described in Vitoratou [22] and reprinted here in the Supporting information (Table in S4 Table).

Page 5, lines 127-130: The 25 statements and the trigger checklist used in this study are reprinted here in the Supporting information (Table in S1 Table) along with the details and examples for the computation of the five factors and the four trigger indices, originally presented in Vitoratou [22, 24].

Page 14, lines 313-315: The scoring guide is presented in Table in S3 Table and the programming codes (SPSS, R project, Stata) to obtain factors and indices are freely available upon request made to the first author.

Page 19, lines 400-401: Moderately discriminative were the S-Five-T scores (Figure in S1 Fig).

---

## [Decision Letter · Decision Letter 1]

23 Feb 2023

Misophonia in the UK: Prevalence and norms from the S-Five in a UK representative sample

PONE-D-22-17376R1

Dear Dr. Vitoratou,

We’re pleased to inform you that your manuscript has been judged scientifically suitable for publication and will be formally accepted for publication once it meets all outstanding technical requirements.

Kind regards,

Sergio A. Useche, Ph.D.

Academic Editor

PLOS ONE

Additional Editor Comments (optional):

Thanks so much for the adjustments made. The paper can now be considered as suitable for publication.

Reviewers' comments:

Reviewer's Responses to Questions

**Comments to the Author**

1. If the authors have adequately addressed your comments raised in a previous round of review and you feel that this manuscript is now acceptable for publication, you may indicate that here to bypass the “Comments to the Author” section, enter your conflict of interest statement in the “Confidential to Editor” section, and submit your "Accept" recommendation.

Reviewer #1: All comments have been addressed

2. Is the manuscript technically sound, and do the data support the conclusions?

Reviewer #1: Yes

3. Has the statistical analysis been performed appropriately and rigorously? 

Reviewer #1: Yes

4. Have the authors made all data underlying the findings in their manuscript fully available?

Reviewer #1: Yes

5. Is the manuscript presented in an intelligible fashion and written in standard English?

Reviewer #1: Yes

6. Review Comments to the Author

Reviewer #1: (No Response)

7. PLOS authors have the option to publish the peer review history of their article (what does this mean?). If published, this will include your full peer review and any attached files.

Reviewer #1: No

---

## [Editor Report · Acceptance letter]

1 Mar 2023

PONE-D-22-17376R1 

Misophonia in the UK: Prevalence and norms from the S-Five in a UK representative sample 

Dear Dr. Vitoratou:

I'm pleased to inform you that your manuscript has been deemed suitable for publication in PLOS ONE. Congratulations! Your manuscript is now with our production department. 

Kind regards, 

on behalf of

Dr. Sergio A. Useche 

Academic Editor

PLOS ONE